# Phytochemically Derived Zingerone Nanoparticles Inhibit Cell Proliferation, Invasion and Metastasis in Human Oral Squamous Cell Carcinoma

**DOI:** 10.3390/biomedicines10020320

**Published:** 2022-01-29

**Authors:** Cheng-Mei Yang, Tian-Huei Chu, Kuo-Wang Tsai, Shuchen Hsieh, Mei-Lang Kung

**Affiliations:** 1Department of Stomatology, Kaohsiung Veterans General Hospital, Kaohsiung 813414, Taiwan; ycmei@vghks.gov.tw; 2Department of Dental Technology, Shu-Zen Junior College of Medicine and Management, Kaohsiung 82144, Taiwan; 3Medical Laboratory, Medical Education and Research Center, Kaohsiung Armed Forces General Hospital, Kaohsiung 80284, Taiwan; y1108020004@mail.802.org.tw; 4Department of Research, Taipei Tzu Chi Hospital, Buddhist Tzu Chi Medical Foundation, New Taipei City 23142, Taiwan; tch33225@tzuchi.com.tw; 5Department of Chemistry, National Sun Yat-sen University, Kaohsiung 80424, Taiwan; shsieh@faculty.nsysu.edu.tw; 6Department of Medical Education and Research, Kaohsiung Veterans General Hospital, Kaohsiung 813414, Taiwan

**Keywords:** phytochemical, zingerone nanoparticle, chemoprevention, metastasis, epithelial–mesenchymal transition, human oral squamous cell carcinoma

## Abstract

Due to its aggressiveness and high mortality rate, oral cancer still represents a tough challenge for current cancer therapeutics. Similar to other carcinomas, cancerous invasion and metastasis are the most important prognostic factors and the main obstacles to therapy for human oral squamous cell carcinoma (OSCC). Fortunately, with the rise of the nanotechnical era and innovative nanomaterial fabrication, nanomaterials are widely used in biomedicine, cancer therapeutics, and chemoprevention. Recently, phytochemical substances have attracted increasing interest as adjuvants to conventional cancer therapy. The ginger phenolic compound zingerone, a multitarget pharmacological and bioactive phytochemical, possesses potent anti-inflammatory, antioxidant, and anticancer activities. In our previous study, we generated phytochemically derived zingerone nanoparticles (NPs), and documented their superior antitumorigenic effect on human hepatoma cells. In the present study, we further investigated the effects of zingerone NPs on inhibiting the invasiveness and metastasis of human OSCC cell lines. Zingerone NPs elicited significant cytotoxicity in three OSCC cell lines compared to zingerone. Moreover, the lower dose of zingerone NPs (25 µM) markedly inhibited colony formation and colony survival by at least five-fold compared to zingerone treatment. Additionally, zingerone NPs significantly attenuated cell motility and invasiveness. In terms of the signaling mechanism, we determined that the zingerone NP-mediated downregulation of Akt signaling played an important role in the inhibition of cell viability and cell motility. Zingerone NPs inhibited matrix metalloproteinase (MMP) activity, which was highly correlated with the attenuation of cell migration and cell invasion. By further detecting the roles of zingerone NPs in epithelial–mesenchymal transition (EMT), we observed that zingerone NPs substantially altered the levels of EMT-related markers by decreasing the levels of the mesenchymal markers, N-cadherin and vimentin, rather than the epithelial proteins, ZO-1 and E-cadherin, compared with zingerone. In conclusion, as novel and efficient phytochemically derived nanoparticles, zingerone NPs may serve as a potent adjuvant to protect against cell invasion and metastasis, which will provide a beneficial strategy for future applications in chemoprevention and conventional therapeutics in OSCC treatment.

## 1. Introduction

The major subtypes of head and neck squamous cell carcinomas (HNSCCs) are derived from the oral cavity and oropharynx. Among them, oral squamous cell carcinoma (OSCC) represents 90% of lip and oral cancers [1]. Moreover, countries with a middle or lower human development index (HDI), such as southern Asia (especially in India and Sri Lanka) and the Pacific Islands (especially in Papua New Guinea), show a higher incidence rate of cancers of the lip and oral cavity. Notably, cancers of the lip and oral cavity are also the leading cause of cancer-related death among men in India and Sri Lanka [2].

Several risk factors have been suggested to be responsible for the etiopathogenesis of oral squamous cell carcinoma (OSCC), including smoking or tobacco consumption, drinking alcohol, chewing areca nut or betel quid [3,4], and infection with human papillomavirus (HPV) [5] or pathogenic and commensal strains of bacteria [6], as well as other associated causes [7,8]. These risks are all involved in chronic inflammation and may lead to the development of oral carcinogenesis. To date, several therapeutic strategies, including surgery, radiation therapy, chemotherapy, and targeted therapy, have been suggested as oral cancer treatments [9]. Nonetheless, the clinical manifestations and the effects of treatment will affect different aspects of the patient’s life. Patients who suffer from oral cancer endure multiple negative psychosocial effects from individuals, families, and health care providers, and physical changes caused by adverse effects on the face, speech, voice, and swallowing, as well as financial impacts that all increase burdens and affect patients’ quality of life [10]. Accordingly, methods to protect against the challenges of tumor metastasis and a poor prognosis have become an important issue for oral cancer therapeutics.

The challenges in curing oral squamous cell carcinoma are the high recurrence rate and lymph node metastasis. Previous studies have indicated that clinical indicators, including the tumor invasion depth [11], bone invasion [12], and vascular and perineural invasion [13], are strongly associated with poor patient prognosis, distant metastasis, and the recurrence of OSCC. Currently, several biological events and mechanisms have been identified and defined in the invasion and metastasis of OSCC cells, including the abnormal regulation of cell adhesion molecules, molecular communication and/or interaction among tumor cells, immune cells, and stromal cells in the tumor microenvironment, the orchestration of cellular signaling pathways, and the involvement of noncoding RNAs [14].

The epithelial–mesenchymal transition (EMT) is a dynamic biological process in which polarized epithelial cells undergo multiple biochemical changes and obtain a mesenchymal phenotype [15]. The EMT plays an important role in normal physiological processes (such as embryogenesis, organ development, tissue regeneration, and organ fibrosis) and pathological processes (including cancer progression and metastasis) [16]. During the EMT, cells reduce the expression of adhesion proteins, such as E-cadherin and ZO-1, and upregulate the expression of cell migratory markers, such as N-cadherin and vimentin [14]. Moreover, cells undergoing EMT increase the proteolytic degradation of the underlying basement membrane by activating matrix metalloproteinases (MMPs), such as MMP2 and MMP9 [17], and activating signal transduction pathways, including PI3K/Akt signaling and TGF-β signaling [18,19]. Therefore, treatments targeting EMT-associated events may be a potential strategy for OSCC therapy.

With the green revolution and the rise of nanoscience, natural compound-based therapeutics, chemoprevention, and health care in human diseases have attracted tremendous amounts of attention due to their potent native bioeffects, including antioxidant, anti-inflammatory, antimicrobial, and anticancer activities [20,21,22]. Phytochemicals, including phenolics, carotenoids, alkaloids, organosulfur compounds, and nitrogen-containing compounds, have been shown to have superior and potent antitumoral activities and interfere with cancer progression. Phytochemicals are therefore regarded as chemopreventive agents against cancer and effective adjuvants during conventional therapeutics [21,23]. However, the higher bioeffective dosage, lower water solubility and bioavailability, and low production are still challenges facing further biomedical development and clinical application. Carbon-based nanoparticles or carbon dots (CDs) are carbonaceous nanoparticles that exhibit unique physicochemical and optical properties analogous to conventional quantum dots and silicon nanoparticles [24]. The attractive characteristics of CDs include their small size (<10 nm), low cost, low cytotoxicity and good biocompatibility, eco-friendliness, abundant functional groups (e.g., amino, hydroxyl, and carboxyl groups), and brightness and high photostability, which all make them good candidates for interdisciplinary applications in bioimaging, biomedicine, and optoelectronic devices [25,26]. In our previous studies, fabricated carbon dots from natural products, including shrimp eggs, herbal essential oil, and phenolic components, displayed superior potential in bioimaging and antibacterial and antitumorigenic activities in vitro and in vivo, respectively [27,28,29].

In this study, we further investigated the efficacy of the as-fabricated, phytochemically derived zingerone nanoparticles (zingerone NPs) in terms of cytotoxicity and inhibiting cell invasion and the EMT in human OSCC cells. Our results broaden the scope of as-prepared, phytochemically derived carbon dots in inhibiting invasion and metastasis, suggesting that they may serve as potent and novel adjuvants in future chemoprevention and cancer therapeutics for oral squamous cell carcinoma.

## 2. Materials and Methods

### 2.1. Cell lines and Nanosized Zingerone

Three human oral squamous cell carcinoma (OSCC) cell lines, including Ca9-22 (the gingival SCC cells), Cal-27, and SAS (the tongue SCC cells), were cultured in DMEM supplemented with 10% fetal bovine serum (Gibco, HyClone, GE Healthcare Life Science, Logan, UT, USA) and 1% penicillin/streptomycin (Primocin, Invivogen, San Diego, CA, USA) [30]. These cell lines were incubated under humidified conditions in 95% air and 5% CO_2_ at 37 °C. Zingerone was purchased from Sigma-Aldrich (Rehovot, Israel) and was used without any purification. Moreover, we generated the zingerone NPs using a hydrothermal methodology that we have described in our previous study [29]. Briefly, zingerone was dissolved in pure ethanol (2% *w/v*) as a stock concentrate of 100 mM. The stock solution was next subjected to heat stirring (1100 rpm) for 4 h at a constant temperature of 120 °C on a hot plate. The obtained zingerone NPs were then stir cooled to room temperature and subjected to filtration using a 0.45 μm polyvinylidene fluoride (PVDF) syringe. Moreover, the obtained zingerone NPs have been analyzed for their uniformity using several experiments, including TEM, photograph images, and UV absorption spectra. Our data demonstrated that the obtained zingerone NPs did not need to be further purified (Appendix A).

### 2.2. Cytotoxicity Assay

Cells (1 × 10^4^ cells/well) were seeded in a 96-well plate overnight and then treated with various concentrations of zingerone and zingerone NPs for 24 h. The effect of zingerone NPs on cell viability was analyzed using a quantitative colorimetric assay with a 3-(4,5-dimethyl thiazol-2-yl)-2,5- diphenyl-tetrazolium bromide (MTT) assay. The cellular formazan was further dissolved with DMSO (Sigma-Aldrich, J.T.Baker, Avantor Performance Materials, LLC., Center Valley, PA, USA), and we determined the optical densities using a microplate reader (Multiskan™ FC Microplate Photometer, Thermo Fisher Scientific, Shanghai, China) at the absorption wavelength of 570 nm.

### 2.3. Colony Formation Assay

Cells (2000 cells/well) were seeded in a 6-well culture plate overnight and incubated with zingerone and/or zingerone NPs for 7–10 days. Cells were then fixed with 4% paraformaldehyde and then stained with crystal violet solution (0.2% *w/v* in 4% paraformaldehyde). The clustered cell clone of over 50 cells was defined as a colony. The colony survival rate was further determined through re-dissolved cellular violet dye with crystal violet elution buffer [29]. A wavelength of 595 nm was used to detect the optical densities using a microplate reader (Multiskan™ FC Microplate Photometer, Thermo Fisher Scientific, Shanghai, China).

### 2.4. Invasion Assay

The invasion assay was performed using a Boyden chamber assay, as previously described [31]. Briefly, the lower chamber was filled with a culture medium (30 µL/well) containing 10% FBS. Following coating a polycarbonate membrane (8 µm pore size Nucleopore) with Matrigel matrix (Corning, Discovery Labware, Inc., Two Oak Park, Bedford, MA, USA), cells (2 × 10^5^ cells/mL) were mixed with various doses of zingerone and/or zingerone NPs and then pipetted into the upper chamber (50 µL/well). By allowing adequate time courses for incubation (28 h for Ca9-22 and SAS cells, and 48 h for Cal-27 cells), the membrane-contained invasive cells were subjected to methanol fixation and Giemsa staining (10% *v/v*, Merck, Sigma-Aldrich Inc., MI, USA). Finally, the invasive cells were counted and photographed using an inverted microscope (ZEISS Axio vert.A1) and counted as mean ± SEM number of cells per filter under six different low-power fields.

### 2.5. Wound Healing Assay

Cell migration was carried out using a commercial Culture-Inserts set (ibidi GmbH, ibidi GmbH; Gräfelfing, Germany). Briefly, cells were seeded in a two-well ibidi Culture-Insert overnight. A cell-free gap was formed by removing the Culture-Insert. Cells were next treated with various doses of zingerone and/or zingerone NPs and incubated for different time intervals. The healing efficiency of the wound gap was monitored using a CYTONOTE 6W image system (iPRASENSE).

### 2.6. Gelatin Zymography Assay

Cells (3 × 10^5^ cells/well) were seeded in a 6-well culture plate and then incubated with 2 mL of serum free medium containing various doses of zingerone and/or zingerone NPs for 24 h. Then, 30 µL of conditioned media was subjected to 10% SDS-PAGE containing 0.1% gelatin (Merck, Millipore, Burlington, MA, USA). Following electrophoresis, the gels were washed with 2.5% Triton X-100 buffer and reacted with activation buffer for 18–24 h, and the gel was then stained with Coomassie blue solution, as previously described [32]. The MMP activity was visualized as clear bands and quantified with a densitometer.

### 2.7. Quantified PCR (qPCR) Analysis

The total RNA from three human OSCC cell lines was extracted using TRIzol reagent (Invitrogen, Carlsbad, CA, USA) and these RNA extracts (2 μg) were further reverse transcribed into first-strand complementary DNA (cDNA) using SuperScript™ III reverse transcriptase (Invitrogen, Carlsbad, CA, USA) by following the manufacturer’s instructions. The mRNA expression levels of *MMP2* were next analyzed using a pair of oligo primers which were described as follows: *MMP2:* Forward, 5′- TGATCTTGACCAGAATACCATCGA-3′; Reverse, 5′- GGCTTGCGAGGGAAGAAGTT-3′. *GADPH*: Forward, 5′-TGCACCACCAACTGCTTAGC-3′; Reverse, 5′-GGCATGGACTGTGGTCATGAG-3′. One twentieth of the complementary DNA generated was used as a template, and gene amplification and detection were achieved using a Fast SYBR^®^ Green PCR Master Mix kit (Applied Biosystems; Life Technologies, Inc., Vilnius, Lithuania) in a StepOnePlus™ Real-Time PCR System (Applied Biosystems; Life Technologies, Inc.).

### 2.8. Western Blot Analysis

The immunoblotting assay was executed as previously described [29]. Briefly, the proteins were separated with 8–12% SDS-PAGE and were transferred onto polyvinylidene fluoride membranes (PVDF). The membrane was then incubated with the primary antibodies of Akt, phospho-Akt, ZO-1 (cell signaling), E-cadherin, N-cadherin, vimentin, and GAPDH (GeneTex, Inc., CA, USA) for 1 h at room temperature or overnight at 4 °C. After the HRP secondary antibody (Santa Cruz Biotechnology, Inc., Heidelberg, Germany) administration, the signals were detected using a Trident femto Western HRP Substrate (GeneTex) and exposed to a ChemiDoc^TM^ XRS Imaging System (BIO-RAD) for autoradiography.

### 2.9. Statistical Analysis

Data are presented as mean ± standard error of the mean (SEM) from indicated repeats of experiments. The statistical analysis was performed using GraphPad Prism 5.0 software (GraphPad Software, San Diego, CA, USA). A *p* value less than 0.05 was considered statistically significant.

## 3. Results

### 3.1. Zingerone NPs Inhibited Proliferation and Suppressed Tumorigenesis in Human Oral Squamous Cell Carcinoma (OSCC) Cell Lines In Vitro

Zingerone is one of the phenolic constituents of *Zingiber officinale*. Moreover, the nonvolatile and pungent compound zingerone possesses multiple pharmacological properties, including anti-inflammatory, anticancer, and antimicrobial properties [33]. In a previous study, we produced zingerone nanoparticles (zingerone NPs), characterized their tetramer structure, which we reconstituted from four zingerone molecules, and further documented their superior antitumorigenic efficacy in human hepatoma [29]. In the present study, we further validated the effect of zingerone NPs on human oral squamous cell carcinoma (OSCC). First, three OSCC cell lines, Ca9-22, Cal-27, and SAS cells, were treated with various doses of zingerone and/or zingerone NPs for 6 h and 24 h. The MTT assay was performed to analyze their cytotoxicity. Our data showed that zingerone NPs slightly affected cell survival. Even in the higher dose of zingerone NP (200 µM) treatment, OSCC cells still maintained 80–100% of cell viability. Moreover, the obtained IC_50_ with either zingerone or zingerone NPs was higher than 330 µM (Appendix A). Therefore, the 24 h treatment has been used for the subsequent experiments. As shown in Figure 1, compared to zingerone, zingerone NPs significantly induced morphological changes in these OSCC cell lines in a dose-dependent manner. For instance, zingerone NPs significantly transformed the regular cell arrangement into a polygon-like phenotype with an altered morphology in Ca-922 cells (Figure 1A). Moreover, zingerone NPs also triggered the occurrence of a loose boundary between cells and a protruded membrane structure in both Cal-27 (Figure 1B) and SAS cells (Figure 1C). Next, zingerone NP-mediated cytotoxicity was also analyzed, and the IC_50_ was calculated and recorded as 118.5 ± 5.6 µM, 174.4 ± 17.5 µM, and 147 ± 17 µM in Ca9-22, Cal-27, and SAS cells, respectively (Figure 1D–F). In addition, based on these results, zingerone NPs elicited substantial cellular stress that reduced survival in these OSCC cell lines compared to raw zingerone treatment.

Next, cells were cultured with zingerone and/or zingerone NPs and then subjected to an in vitro colony formation assay to characterize the effect of zingerone NPs on the tumorigenicity of OSCC cells. As shown in Figure 2, compared to zingerone, zingerone NPs induced a superior inhibition of colony formation in these OSCC cell lines. In Ca9-22 cells, zingerone significantly inhibited colony formation in a dose-dependent manner. Intriguingly, even the lowest dose (10 µM) of zingerone NP treatment exhibited an obvious suppressive efficacy (Figure 2A). Similar results were also obtained using the Cal-27 (Figure 2B) and SAS cell lines (Figure 2C), although the effective dose was higher than that in Ca9-22 cells (approximately 25 µM). Moreover, the colony numbers and colony survival were further analyzed and recorded. Zingerone significantly reduced colony numbers and colony survival in these OSCC cell lines in a dose-dependent manner. Undoubtedly, zingerone NPs elicited more substantial decreases in colony numbers and colony survival. Even at the lowest dose (10 µM), zingerone NPs significantly decreased colony numbers by greater than 60% and colony viability by 70%, and the higher dose (over 25 µM) treatment induced approximately a 100% inhibition of colony survival in Ca9-22 cells (Figure 2D). A similar outcome was also observed in Cal-27 cells (Figure 2E) and SAS cell lines (Figure 2F), indicating that a lower dose of treatment (25 µM) dramatically decreased colony formation and colony viability by greater than 80%. The complete inhibition of colony activity was observed in cells treated with doses greater than 50 µM. Therefore, zingerone NPs inhibited the proliferation and tumorigenicity of OSCC cell lines.

### 3.2. Zingerone NPs Attenuated the Migration and Invasion of Human OSCC Cell Lines

Cell motility and mobility are highly correlated with cell invasion, metastasis, and tumor progression. In OSCC, cellular metastasis into cervical lymph nodes or distant organs is suggested as the most important prognostic indicator of the clinical diagnosis [11,14]. Here, we investigated the effects of zingerone NPs on cell migration and invasion. As shown in Figure 3, zingerone NPs resulted in limited gap closure in all three OSCC cell lines compared to the zingerone group. In Ca9-22 cells, the zingerone groups showed favorable gap-healing efficiency compared to the zingerone NP groups at 15 h. At 24 h, in addition to the highest dose of zingerone (200 µM) and the zingerone NP groups, the wound gaps among the control and the remaining zingerone groups (25, 50, and 100 µM) were all filled. Furthermore, the quantitative data also revealed that the zingerone NP groups elicited good inhibition of cell motility and cell migration. For example, after treatment with the lower dose (50 µM) for 15 h, the gap healing efficiency among the control, zingerone, and zingerone NP groups reached 100%, 85%, and 40%, respectively. Even after 24 h of treatment, approximately 50% of the unhealed gap was still observed in the zingerone NP group compared to the other two groups (Figure 3A). A similar result was also observed in Cal-27 cells, although the healing time of the control group was prolonged to 40 h (Figure 3B). In addition, the zingerone NP-mediated inhibition of SAS cell migration was observed after treatment with higher doses (100 µM) for 15 h (Figure 3C). Altogether, these results suggested that zingerone NPs significantly attenuated cell motility and cell migration.

Next, the effect of zingerone NPs on cell invasion was also examined. Cells were mixed well with zingerone and/or zingerone NPs and then plated in a Boyden chamber equipped with a Matrigel-coated membrane. After an adequate incubation time, cellular invasion was analyzed and recorded. Compared to zingerone, zingerone NPs dramatically inhibited the invasion of these OSCC cell lines in a dose-dependent manner (Figure 4A–C). In Ca9-22 cells, zingerone NPs inhibited invasion by 20–60% in a dose-dependent manner compared to the zingerone group, which did not reach over 20% inhibition. Even the lowest dose (25 µM) of zingerone NP treatment induced 20% inhibition, which was only achieved with the highest dose (200 µM) of zingerone treatment (Figure 4D). In Cal-27 cells, zingerone NPs noticeably decreased invasion by 50–95% in a dose-dependent manner compared to zingerone, which did not substantially affect cell invasion (Figure 4E). In SAS cells, zingerone NPs inhibited invasion to a similar extent as that in Ca9-22 cells (Figure 4F). These results suggested that zingerone NPs exhibited a superior ability to inhibit the invasion of human OSCC cells.

### 3.3. Downregulation of Akt Signaling and EMT Signaling Was Involved in Zingerone NP-Mediated Inhibition of Cell Proliferation and Metastasis

Akt-mediated cell motility and cell invasion through the activation of EMT play important roles in OSCC tumorigenesis and metastasis [19]. Here, we verified the effects of zingerone NPs on Akt signaling. After treatment with various doses of zingerone and/or zingerone NPs for 24 h, cells were harvested and subjected to Western blotting. Zingerone NPs significantly decreased Akt activity in these three OSCC cell lines in a dose-dependent manner (Figure 5A–C). Moreover, the effective dose of zingerone NPs that inhibited Akt activation (approximately 25 µM) was lower than that of zingerone (Figure 5D–F). This result suggested that the zingerone NP-mediated inhibition of Akt signaling played a pivotal role in inhibiting cell proliferation and motility. PI3K/Akt signaling plays critical roles in various pathophysiological processes, including cell proliferation, motility, invasion, and metastasis. PI3K/Akt signaling is thus regarded as a main target for the development of cancer therapeutics [34]. Indeed, a previous study suggested that therapies for human tongue carcinomas based on Akt inhibition might be a strategy for controlling the Akt-driven EMT and cell motility and invasiveness [19]. Accordingly, the phytochemical zingerone NP-mediated inhibition of Akt activity to abolish cell proliferation and motility and invasion in OSCC may represent a novel candidate chemopreventive adjuvant for OSCC treatment.

Overcoming the extracellular matrix (ECM) barrier is the first step in tumor cell invasion and distant metastasis. The enzymatic digestion of ECM by matrix metalloproteinases (MMPs) is an important component of normal physiological and pathological processes, such as embryonic development, wound healing, inflammation, and cancer [35]. In addition, due to the critical roles of MMPs in tumor cell invasion and metastasis, MMPs such as MMP2 and MMP9 have been identified as prognostic factors for many types of cancer, such as head and neck squamous cell carcinomas (HNSCCs) and oral cancers. [36,37] Therefore, we further analyzed the effects of zingerone NPs on MMP activity. Using a zymography assay, we found that zingerone NPs significantly attenuated MMP2 and MMP9 activity in these three OSCC cell lines compared to zingerone treatment (Figure 6A–C). Moreover, the effect of zingerone NPs on reducing the levels of lower molecular weight MMP proteins (~45–50 kDa) was also observed (Appendix A). Additionally, compared to MMP9, zingerone NPs seemed to be more effective at decreasing MMP-2 activity in a dose-dependent manner. Accordingly, we further analyzed the expression level of the MMP2 mRNA using quantitative PCR analysis. In Ca9-22 cells, zingerone significantly increased MMP2 mRNA levels. Conversely, zingerone NPs dramatically suppressed MMP2 mRNA expression in a dose-dependent manner, and the effective dose for MMP2 mRNA inhibition was observed with the 50 µM treatment (Figure 6D). A similar result was also observed in Cal-27 and SAS cells; in particular, a greater decrease in MMP2 mRNA levels was observed after treatment with the higher dose (100 µM) of zingerone NPs (Figure 6E,F). According to these data, zingerone NPs abolished cell migration and invasion, and may simultaneously decrease the activities of various MMPs, especially MMP2 and MMP9. Indeed, several MMPs have been shown to digest gelatin, such as MMP1, MMP3, MMP10, and MMP12, whose molecular weights range from approximately 54 to 57/44 to 45 kDa in latent/active forms, respectively [38]. Moreover, these MMPs also play roles in various types of cancer and metastasis [39].

On the other hand, due to the superior efficacy of zingerone NPs in suppressing cellular invasion, we next investigated the effects of zingerone NPs on the levels of EMT-related markers, such as the epithelial markers ZO-1 and E-cadherin, and the mesenchymal markers N-cadherin and vimentin. In Ca9-22 cells, zingerone NPs significantly reduced the levels of the mesenchymal markers of N-cadherin and vimentin, rather than the epithelial marker of ZO-1, in a dose-dependent manner compared with zingerone. Moreover, neither zingerone nor zingerone NPs induced an obvious change in the epithelial marker of E-cadherin protein levels (Figure 7A,B). Interestingly, in Cal-27 cells, zingerone achieved maximal reductions of 30% and 45% in the levels of epithelial markers (ZO-1 and E-cadherin) and mesenchymal markers (vimentin, but not N-cadherin), respectively. In contrast to zingerone, zingerone NP treatment induced substantial decreases in the levels of both epithelial markers and mesenchymal markers in a dose-dependent manner (the maximal inhibition achieved was >90%) (Figure 7C,D). Moreover, in SAS cells, zingerone treatment induced a slight increase in the levels of mesenchymal markers (N-cadherin and vimentin) compared with epithelial markers (E-cadherin, but not ZO-1). Conversely, zingerone NPs significantly reduced the levels of both epithelial markers and mesenchymal markers, and the effective dose was at least 100 µM, which achieved a decrease of approximately 60% (Figure 7E,F). These results revealed that zingerone NPs substantially altered the levels of EMT-related markers, and especially interfered with mesenchymal markers compared to epithelial markers among these OSCC cell lines.

## 4. Discussion

Previous studies have indicated that zingerone exhibits different cytotoxic effects on different cancer cell types. For example, a lower dose (IC_50_ < 25 µM) of zingerone treatment has been shown to induce >50% cytotoxicity in both colon cancer and prostate cancer [40,41], while a high dose (IC_50_ > 2000 µM) of zingerone treatment merely elicited 40% cytotoxicity in neuroblastoma cells [42]. Indeed, higher tolerance to zingerone treatment (IC_50_ is over 200 µM) was also observed in OSCC cells and hepatoma cells [29]. Here, we fabricated green and carbon-based zingerone NPs and showed that they possessed superior cytotoxicity and antitumorigenic activity. For instance, a lower dose (25 µM) of zingerone NP treatment inhibited tumorigenesis by 50% and 80% in hepatoma cells [29] and OSCC cells, respectively, whereas higher doses of zingerone (>200 µM) may be needed to achieve a similar efficacy. Notably, the safety of zingerone NPs was also validated in our previous study. After mice were fed either zingerone or zingerone NPs for 8 weeks (0.4 mg/25 g BW), neither treatment elicited significant liver toxicity (based on the GOT/GPT index of the sera) [29]. Moreover, in vitro data also validated that the normal bone marrow HS-5 cells had higher tolerance to zingerone NP treatment (IC_50_ = 184.2 ± 6.8 µM) compared to these OSCC cell lines (Appendix A). This result suggested that zingerone NPs revealed good biocompatibility and safety toward normal cells.

The as-prepared zingerone NPs had an ultrasmall size of 1.42 ± 0.67 nm and were clear with an amber color. Moreover, zingerone NPs have a negative charge (−15.99 ± 0.23 mV), and an absorption peak at 350 nm is observed in the UV–vis spectrum due to the π−π* transition of the benzenoid rings. The total ion chromatogram analysis also reveals functional groups, including methoxy and hydroxyl groups, on the nanoparticle. Alterations in the physicochemical properties may increase the pharmacological efficacy of zingerone NPs in cancer cells. Indeed, nanosized zingerone may have more surface chemistry consisting of methoxy and hydroxyl groups than zingerone alone [29]. Moreover, methoxy groups and hydroxyl groups have been reported play pivotal roles in the bioactivities of natural compounds, such as anticancer, antimigration, and antiproliferative activities [43]. Furthermore, unlike positively charged nanoparticles, which are internalized by cells through an electrostatic interaction between cationic NPs and the negatively charged cell membrane, negatively charged nanoparticles adsorb to the nuclear membrane and interact with nuclei, potentially because the nucleolar pH is consistently 0.3 to 0.5 units higher than that of the cytosol [44,45,46]. Therefore, variations in physicochemical properties, such as the ultrasmall size, shape, surface charge, and surface chemistry, may increase the uptake of nanoparticles by cells and allow them to interact with macromolecules, such as nucleotides, lipids, and proteins, thus interfering with cellular biofunctions [47,48]. These outcomes may ultimately induce lethal stress in cancer cells. On the other hand, we have reported that zingerone NPs significantly increased cell apoptosis, decreased cell proliferation and disturbed the cell cycle distribution in human hepatoma cells. These results are attributed to the zingerone NP-mediated downregulation of Akt/NF-κB signaling, increased DNA damage and DNA instability, and the activation of caspase-mediated apoptosis signaling pathways. These outcomes finally contribute to the inhibitory effects of zingerone NPs on the proliferation and tumorigenicity of hepatoma cells [29]. Accordingly, both Akt/NF-κB signaling and the caspase signaling cascade might be involved in the zingerone NP-mediated inhibitory effects on proliferation and tumorigenicity in OSCC cell lines.

Recently, zingerone has attracted increasing interest from researchers for its chemopreventive potential, including eliciting antiangiogenic activity to protect against tumor development [49], protecting against colon carcinogenesis mediated by the carcinogen 1,2-dimethylhydrazine (DMH) in rats [50], and inducing the mitotic arrest of the cell cycle to suppress neuroblastoma development [42]. However, a relatively higher dose is necessary to achieve therapeutic efficacy. To date, compared to the raw phytochemical compounds, phytochemically derived carbon dots (CDs) have been reported to have superior efficacy in biomedical applications, including antiviral activity and anticancer efficiency. For instance, curcumin-derived carbon quantum dots (Cur-CQDs) significantly inhibited enterovirus 71 (EV71) infection (half-maximal effective concentration (EC_50_) < 20 µg/mL compared to curcumin (EC_50_ > 200 µg/mL)) and protected newborn mice from virus-induced hindlimb paralysis [51]. Moreover, berberine is an isoquinoline alkaloid that possesses good anticancer, antibacterial, and anti-inflammatory bioactivities [52]. A previous study reported that berberine-based carbon dots (Ber-CDs) not only had superior optical properties for bioimaging and retained the biofunctions of berberine, but also exhibited selective and safe antitumor performance (with a decrease of approximately 50% of tumor nodules) [53]. In the present study, a lower dose (<25 µM) of zingerone-based carbon dots (zingerone NPs) not only elicited significant antiproliferative and antitumor activity, but also inhibited the migration and invasion of OSCC cells.

Cancer cells that metastasize into the cervical lymph nodes and/or distant organs have become the most important prognostic indicator of oral squamous cell carcinoma (OSCC). Moreover, several signaling mechanisms have been identified that are involved in OSCC invasion and metastasis, such as MMP activity, cadherin switching-associated EMT-related signaling cascades, and signaling pathways, such as the PI3K/Akt signaling, TGF-β, and Wnt/β-catenin pathways [54]. Based on accumulating evidence, phytochemicals interfere with various intracellular mechanisms in many types of cancer, including the induction of apoptosis, the arrest of cell cycle progression, the blockade of transcription factors, such as NF-κB, the induction of oxidative stress, the suppression of MMP activity, and the inhibition of signal transduction pathways, such as the Akt, VEGF, Wnt, and STAT3 signaling pathways [21,55]. In addition, as potent adjuvants, accumulating studies have suggested that phytochemicals significantly improve cancer invasion and metastasis, and attenuate side effects and drug resistance during treatment with conventional tumor therapeutics [56,57].

Additionally, compared to the tongue squamous cell carcinoma cell lines, Cal-27 and SAS, zingerone NPs showed greater bioactivity and suppression of gingival squamous cell carcinoma Ca9-22 cells activity in the MTT assay (Figure 1), colony formation assay (Figure 2), wound healing assay (Figure 3), and MMP activity assay (Figure 6). Interestingly, by analyzing the EMT-related markers, zingerone NPs obviously reduced the levels of epithelial markers in the Cal-27 and SAS cell lines rather than in Ca9-22 cells (Figure 7). Zingerone NPs also exerted different inhibitory effects on invasion (Figure 4) and Akt activity (Figure 5) among these OSCC cell lines. Although the Ca9-22, Cal-27, and SAS cell lines were classified as aggressive OSCC cells in the cancer grading system, differences in the genome and/or gene diversity, and variations in the cellular molecular biology, may guide different cellular responses to environmental stimuli. For example, Ca9-22 cells, a malignant gingival squamous cell carcinoma line that was extracted from a 43-year-old Japanese male, were identified as tetraploid cells without the typical normal chromosome form 46XY [58]. Cal-27 cells are a form of malignant tongue squamous cell carcinoma that were extracted from a 56 year-old Caucasian male, and a cytogenetic analysis revealed moderate hyperploidy; the average number of chromosomes per cell analyzed was 43. One of the chromosomes of certain pairs was always (pairs 8, 9, 11, 13, 22) or almost always (pairs 12, 21) missing [59]. The SAS cells, representing poorly differentiated tongue squamous cell carcinoma, were extracted from a 69-year-old Japanese female [60]. All the different genomic backgrounds may contribute to the differences in tolerance to zingerone and/or zingerone NP treatment. Moreover, the gene expression profile and differences in genetic heterogeneity in different races and sexes would also confer differences in sensitivity to zingerone and/or zingerone NP treatment.

Altogether, we combined both phytochemicals and nanotechnology to enhance raw zingerone bioactivity, and attempted to develop zingerone NPs as potent nanomedicines and/or phytochemical nano-adjuvants for not only hepatocellular carcinoma cells [29], but also oral squamous carcinoma cells. In this study, the as-prepared zingerone NPs exhibited impressive efficacy in decreasing levels of cell invasion-associated factors, such as MMPs and Akt signaling, and interfered with EMT-associated proteins, which all led to reduced cell survival and inhibited the invasion and metastasis of OSCC cells (Figure 8).

## 5. Conclusions

Nanoscience is rapidly reorganizing the landscapes of various cancer therapeutic strategies and chemopreventive agents. In the present study, we fabricated phytochemically derived zingerone NPs and documented their superior bioactivity and multiple pharmacological effects on OSCC cells, including their inhibitory effects on proliferation, motility, invasion, and tumorigenesis. Our data indicate that zingerone NPs significantly suppress the activities of the Akt protein and MMP enzymes, decrease MMP2 mRNA expression, and disturb EMT marker expression. Zingerone NPs thus exert potent inhibitory effects on the invasion and metastasis of OSCC cells.

Altogether, these results suggest that zingerone NPs may represent a potent chemopreventive adjuvant and provide an alternative strategy for treating OSCC that improves the side effects, drug resistance, and recurrence associated with conventional cancer therapies.

## Figures and Tables

**Figure 1 biomedicines-10-00320-f001:**
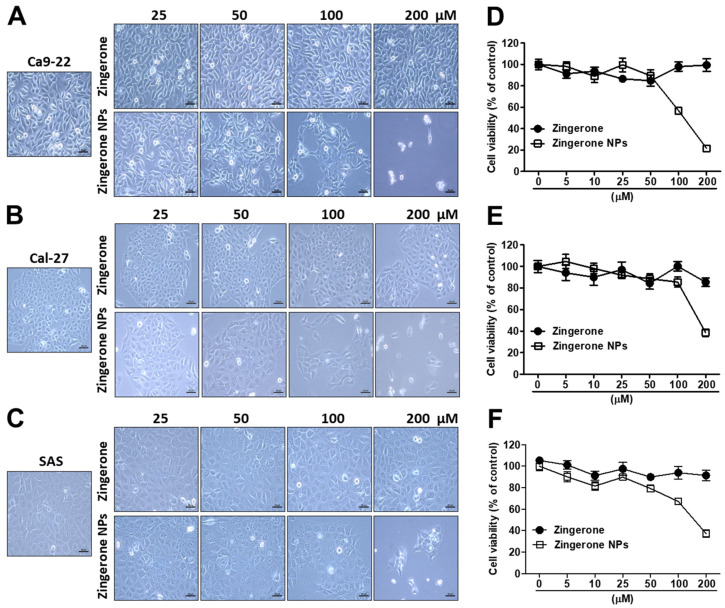
Zingerone NPs induced a significant cytotoxicity in three human OSCC Ca9-22, Cal-27 and SAS cell lines. Cells were treated with zingerone and/or zingerone NPs for 24 h. The effects of zingerone and/or zingerone NPs on cellular morphology and cell viability were observed in Ca9-22 cells (**A**), Ca1-27 cells (**B**), and SAS cells (**C**), respectively, using inverted microscopy. Bar: 50 µm. Further cellular viability was analyzed and recorded using an MTT assay in Ca9-22 cells (**D**), Ca1-27 cells (**E**), and SAS cells (**F**), respectively. Data are expressed as mean ± SEM of three experiments.

**Figure 2 biomedicines-10-00320-f002:**
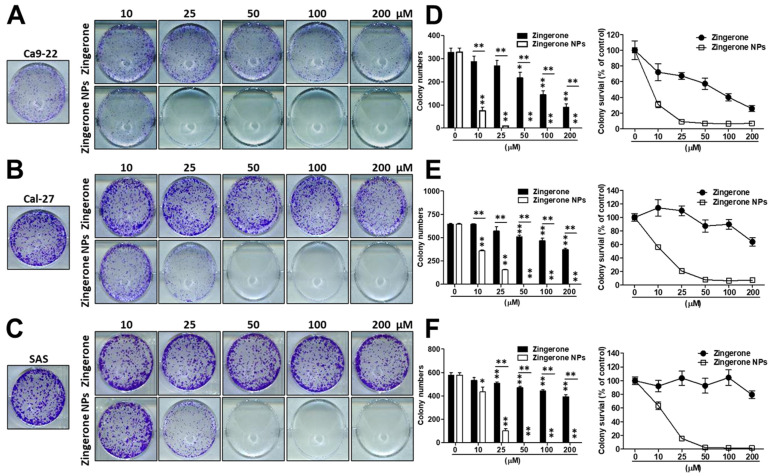
Zingerone NPs elicited a superior anti-tumorigenicity in three human OSCC Ca9-22, Cal-27, and SAS cell lines in vitro. Cells were treated with various doses of zingerone and/or zingerone NPs for 7–10 days. The colony formation was further stained and detected using crystal violet staining. The zingerone NP-mediated anti-tumorgenicity on Ca9-22 cells (**A**), Ca1-27 cells (**B**), and SAS cells (**C**) was observed and recorded using inverted microscopy. Quantitative analysis of colony formation was performed, and either colony numbers or colony survival rate for (**D**) Ca9-22 cells, (**E**) Ca1-27 cells, and (**F**) SAS cells, respectively, were further counted and determined using absorbance detection at 595 nm. All data are expressed as the mean ± SEM of three experiments. * *p* < 0.05, ** *p* < 0.01.

**Figure 3 biomedicines-10-00320-f003:**
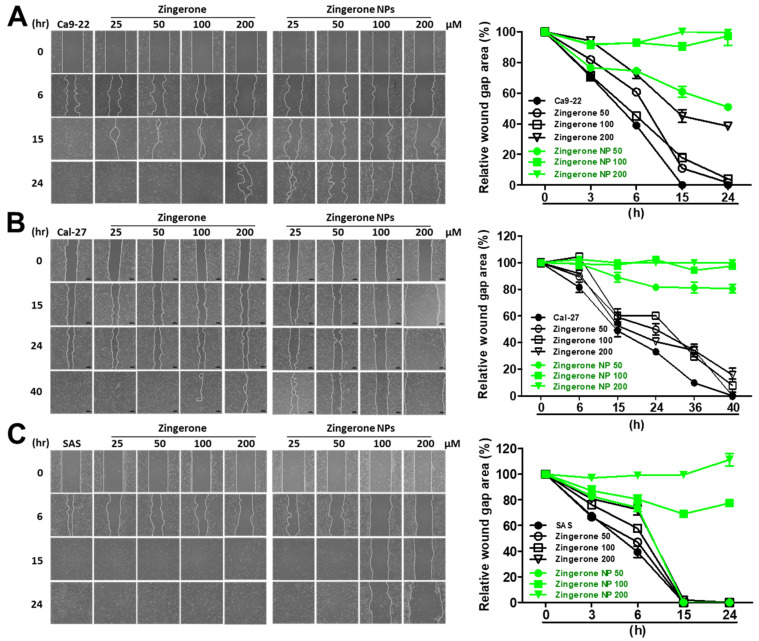
Effect of nanosized zingerone on cell motility of human OSCC cell lines**.** Dose and time effects of zingerone NPs on (**A**) Ca9-22 cells, (**B**) Ca1-27 cells, and (**C**) SAS cells were validated and imaged using wound scratch assays. The percentages of gap healing were further analyzed and quantified at the indicated time intervals (right panels). Data are mean ± SEM of at least three independent experiments.

**Figure 4 biomedicines-10-00320-f004:**
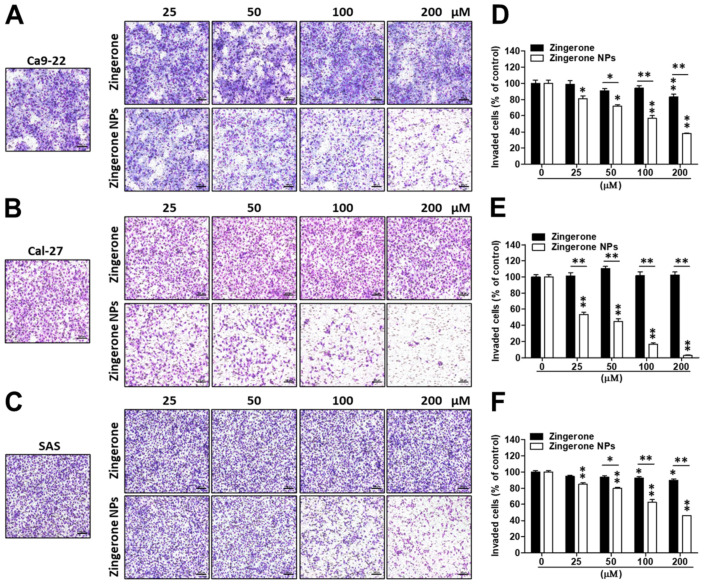
Effect of nanosized zingerone on cell invasion of three human OSCC cell lines. Cells were treated with various doses of zingerone and/or zingerone NPs and then subjected to Boyden chamber assays. Following incubation for an adequate time, the cell invasion of three human OSCC cell lines, (**A**) Ca9-22 cells, (**B**) Ca1-27 cells, and (**C**) SAS cells, was imaged and recorded using inverted microscopy. Bar: 100 µm. The invasive cells of (**D**) Ca9-22 cells, (**E**) Ca1-27 cells, and (**F**) SAS cells were further counted and quantified. Bars represented as mean ± SEM for n = 5 membranes in three separate experiments (* *p* < 0.05; ** *p* < 0.01).

**Figure 5 biomedicines-10-00320-f005:**
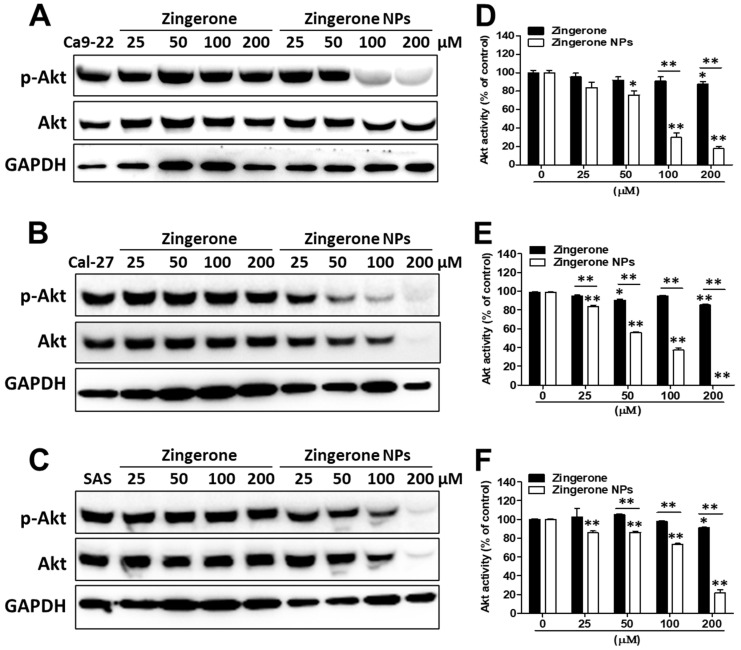
Effect of zingerone NPs on Akt activity of human OSCC cell lines. Cells were treated with zingerone and/or zingerone NPs for 24 h. The Akt activity was detected on (**A**) Ca9-22 cells, (**B**) Ca1-27 cells, and (**C**) SAS cells, respectively using Western blotting. The Akt activity was further analyzed and quantitated for (**D**) Ca9-22 cells, (**E**) Ca1-27 cells, and (**F**) SAS cells. Data are expressed as mean ± SEM of three experiments. * *p* < 0.05, ** *p* < 0.01.

**Figure 6 biomedicines-10-00320-f006:**
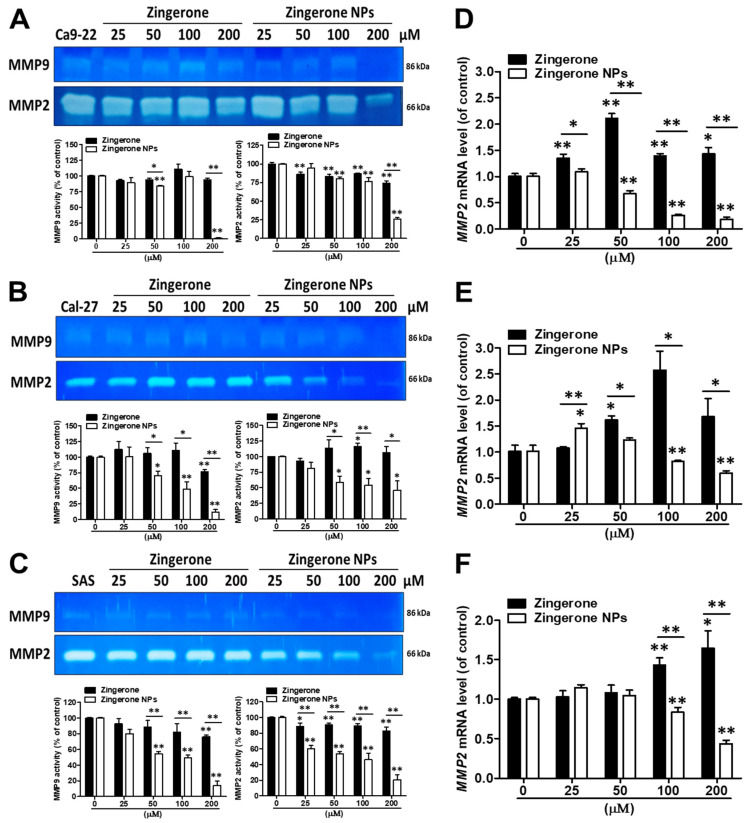
Effects of zingerone NPs on MMPs activity of human OSCC cell lines. Cells were incubated in serum-free DMEM and contained with various doses of zingerone and/or zingerone NPs for 24 h. The conditioned media of (**A**) Ca9-22 cells, (**B**) Ca1-27 cells, and (**C**) SAS cells were collected to analyze MMP activity, including MMP2 and MMP9, using a gelatin zymography assay. Moreover, the mRNA expression levels of MMP2 in (**D**) Ca9-22 cells, (**E**) Ca1-27 cells, and (**F**) SAS cells were analyzed using qPCR analysis. Data are mean ± SEM of three independent experiments. * *p* < 0.05, ** *p* < 0.01.

**Figure 7 biomedicines-10-00320-f007:**
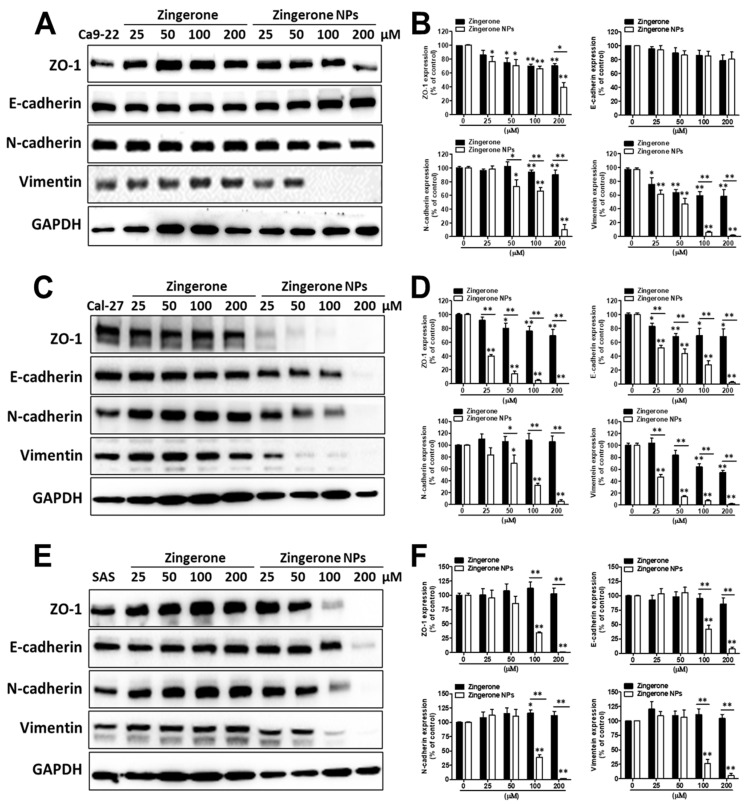
Zingerone NPs disturbed the expression levels of EMT-associated proteins in human OSCC cell lines. Cells were treated with zingerone and/or zingerone NPs for 24 h. The EMT-associated proteins, including ZO-1, E-cadherin, N-cadherin, and vimentin, were analyzed and quantitated on Ca9-22 cells (**A**,**D**), Ca1-27 cells (**B**,**E**), and SAS cells (**C**,**F**), respectively, using Western blotting assay. Data are expressed as mean ± SEM of three experiments. * *p* < 0.05, ** *p* < 0.01.

**Figure 8 biomedicines-10-00320-f008:**
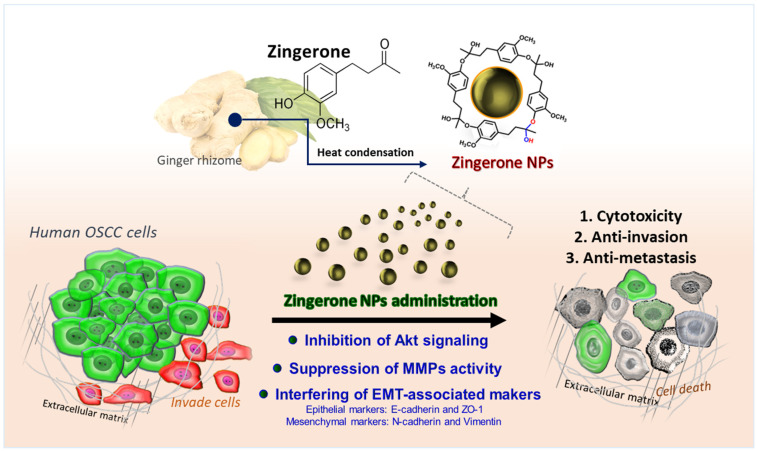
The phytochemically derived zingerone NPs elicited superior efficacy to inhibit the cell invasion and metastasis of human oral squamous cell carcinoma. The as-fabricated zingerone NPs significantly suppressed Akt signaling-mediated cell survival and cell motility, which led to obvious cytotoxicity and anti-proliferation. Moreover, the zingerone NP-mediated downregulation of MMP activity was also reflected in the harsh cell motility. In addition, the zingerone NPs substantially disturbed the expression levels of EMT-associated markers, including the mesenchymal markers N-cadherin and vimentin, and the epithelial markers ZO-1 and E-cadherin. These results suggested that the zingerone NPs exerted superior suppression on cell proliferation, tumorigenicity, and cell motility, and thus achieved an inhibitory effect against the cell invasion and metastasis of human oral squamous cell carcinoma.

## Data Availability

Not applicable.

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
