# Peer review of "Phytochemically Derived Zingerone Nanoparticles Inhibit Cell Proliferation, Invasion and Metastasis in Human Oral Squamous Cell Carcinoma"

_biomedicines, 2022, doi:10.3390/biomedicines10020320_

Round 1
Reviewer 1 Report
The presented manuscript represents a continue of the paper called “Zingerone Nanotetramer Strengthened the Polypharmacological Efficacy of Zingerone on Human Hepatoma Cell Lines”. Zingerone NPs, previously exhibiting good efficiency in treatment of human malignant hepatoma cells, were now tested against oral squamous cell carcinoma. The results have shown that zingerone NPs significantly decrease cell viability, cell motility and invasiveness of three investigated OSCC cell lines. Observed results are ascribed to the downregulation of Akt-signalling in activation of EMT-transition – same mechanism as in the case of previously investigated hepatoma cells.
The results of the investigation are interesting and significant, particularly these considering the favorable effects on cell invasiveness and metastasis. However, more attention should be paid to the comparison and discussion in light of the previously observed effects on human hepatoma cells (differences and similarities), due to the same mechanism of action proposed. Also, the observed better efficiency of zingerone NPs compared to pure zingerone is the key part of the paper, and requires more detailed explanation from the physico-chemical point of view. Is there any possibility to propose and identify the exact binding sites of zingerone and zingerone NPs in the proteins in the affected pathways? It would be very interesting to compare the effects of zingerone NPs to pure zingerone on the site of action, and it certainly would contribute to the novelty of the paper.
Author Response
Response to Reviewers Comments
Reviewer #1:
The presented manuscript represents a continue of the paper called “Zingerone Nanotetramer Strengthened the Polypharmacological Efficacy of Zingerone on Human Hepatoma Cell Lines”. Zingerone NPs, previously exhibiting good efficiency in treatment of human malignant hepatoma cells, were now tested against oral squamous cell carcinoma. The results have shown that zingerone NPs significantly decrease cell viability, cell motility and invasiveness of three investigated OSCC cell lines. Observed results are ascribed to the downregulation of Akt-signalling in activation of EMT-transition – same mechanism as in the case of previously investigated hepatoma cells. The results of the investigation are interesting and significant, particularly these considering the favorable effects on cell invasiveness and metastasis.
- However, more attention should be paid to the comparison and discussion in light of the previously observed effects on human hepatoma cells (differences and similarities), due to the same mechanism of action proposed. Also, the observed better efficiency of zingerone NPs compared to pure zingerone is the key part of the paper, and requires more detailed explanation from the physico-chemical point of view.
Response: We appreciate the reviewer’s comments. We have added a detailed description of the physicochemical properties for zingerone NPs in the resubmitted manuscript. As shown below:
The page 9 with yellow mark in the in the “Discussion” section
The as-prepared zingerone NP is an ultrasmall size of 1.42 ± 0.67 nm and owns a clear and amber apparent. Moreover, the zingerone NP possesses a negative charge (−15.99 ± 0.23 mV) and is observed an absorption peak at 350 nm of the UV-vis spectra due to π−π* transition of the benzenoid rings. The total ion chromatogram analysis also elucidates an exhibition of functional groups including methoxy and hydroxyl groups on the nanoparticle.……Indeed, the nanosized zingerone may own more surface chemistry exhibition of methoxy and hydroxyl groups rather than zingerone alone.[29]
- Is there any possibility to propose and identify the exact binding sites of zingerone and zingerone NPs in the proteins in the affected pathways? It would be very interesting to compare the effects of zingerone NPs to pure zingerone on the site of action, and it certainly would contribute to the novelty of the paper.
Response: We appreciate the reviewer’s keen observation and excellent point. However, due to insufficient laboratory staff and funding constraints, it is currently impossible to implement this issue. We will include the issue in our future works to identify the exact binding sites of zingerone and zingerone NPs in the proteins in the affected pathways. Nonetheless, so far, the accumulating pieces of knowledge toward the understanding of zingerone NP on cellular signaling are including the downregulation of Akt/NFkB signaling, enhanced DNA damage and DNA instability, and triggered the activation of caspases-mediated apoptosis signaling in hepatoma cells. Moreover, zingerone NP-mediated the downregulation of the MMPs activity and Akt signaling and interfered with the expression profiles of EMT-associated markers in OSCC cells were also observed. In addition, we also found that zingerone NP significantly disturbed cytoskeleton reorganization in melanoma cells which was attributed to the dysregulation of Src/FAK signaling and some signaling involved in cell cycle progression in colon cancer cells (data not published) and so on. Accordingly, due to the physicochemical properties change, we do believe that zingerone NP may comprehensively interact with proteins, nucleotides, and enzymes that finally contributed to anti-cell proliferation, induction of cell apoptosis, anti-invasion, and anti-tumorigenesis.

Reviewer 2 Report
The paper of Yang et al. is well written and comprise interesting data. The conclusions are well supperted with data and some of them are provided as supplementary data.
(1) I think that the molecular structure of zingerone should be placed to the introduction.
(2) Authors should check once the formatting of the text.
(3) Authors should correct the formatting of the citations.
Overall the paper is suitable for publication in Biomedicines MDPI journal after minor revisions.
Author Response
Response to Reviewers Comments
Reviewer #2:
Comments and Suggestions for Authors
The paper of Yang et al. is well written and comprise interesting data. The conclusions are well supperted with data and some of them are provided as supplementary data.
- I think that the molecular structure of zingerone should be placed to the introduction.
Response: Thanks for the reviewer’s suggestion. The molecular structure of zingerone has been shown in Figure 8 of the resubmitted manuscript.
- Authors should check once the formatting of the text.
Response: We appreciate the reviewer’s comments. We have checked and reorganized the formatting of the text.
- Authors should correct the formatting of the citations.
Response: We appreciate the reviewer’s comments. We have checked and corrected the formatting of the citations.
- Overall, the paper is suitable for publication in Biomedicines MDPI journal after minor revisions.
Response: We appreciate the reviewer’s comments.

Reviewer 3 Report
In this work, Yang and co-workers evaluate the therapeutic potential of zingerone and zingerone-based nanoparticles toward different human oral squamous cell carcinomas.
This topic of research is well within the scope of this journal, and should be of high interest for its readership. The obtained nanoparticles did show some therapeutic potential, which was relatively well characterized by the authors. The manuscript is also relatively well-written and easy to follow. However, the manuscript still presents some flaws that need to be addressed in major revision. Namely:
-I think that the abstract would benefit from being shorter and more concise;
-Why the focus in the Introduction section on Carbon dots, and what is its relationship with the present study?
-Why the incubation time of 24h for the cytotoxicity assays? It would be interesting to study higher incubation times and see their potential effects on the cytotoxicity of the nanoparticles;
-While the studied nanoparticles were already reported, it would beneficial for the readers if the authors describe them, their synthesis and properties in more detail in the present paper;
-The authors should compare the cytotoxicity/IC50 values obtained for both zingerone and zingerone NPs for reference chemotherapeutic drugs for this type of cancer. Such information is essential to really understand the therapeutic potency of the reported nanoparticles;
-Use of adjectives as "excellent" is not very scientific, and so, the authors should refrain from using them;
-The zingerone nanoparticles showed different activity than that of molecular zingerone. The authors should discuss in more detail potential reasons for that;
-The authors should also discuss reasons for potential differences between obtained results for different cell lines;
-The authors obtained results for carcinoma cells. To assess potential biocompatibility/safety issues, the authors should also evaluate the present nanoparticles toward non-carcinoma cells.
-What is the synthesis yield (mass per mass, in %) of the obtained nanoparticles?
-How many replicate synthesis were performed for the nanoparticles, to ensure that they presented reproducible properties?
-To my understanding, the authors present these nanoparticles as carbon dot-like, which were obtained by bottom-up procedures. It should be noted that bottom-up synthesis of carbon dots produce also molecular impurities, besides the dots, which can only be removed from solution by either dialysis or column chromatography (see DOIs: 10.1021/acs.chemmater.7b04446; 10.1039/C9CP03730F). However, as I understood from their ACS Appl Mater Interface paper, the authors did neither. So, the authors should discuss how they guarantee the absence of potential molecular impurities;
-The authors should discuss how they can guarantee the uniformity of the population of obtained nanoparticles, without further purification;
Author Response
Response to Reviewers Comments
Reviewer #3:
Comments and Suggestions for Authors
In this work, Yang and co-workers evaluate the therapeutic potential of zingerone and zingerone-based nanoparticles toward different human oral squamous cell carcinomas. This topic of research is well within the scope of this journal, and should be of high interest for its readership. The obtained nanoparticles did show some therapeutic potential, which was relatively well characterized by the authors. The manuscript is also relatively well-written and easy to follow. However, the manuscript still presents some flaws that need to be addressed in major revision. Namely:
- I think that the abstract would benefit from being shorter and more concise.
Response: Thanks for the reviewer’s suggestion. We have shortened the abstract and made it more concise in this resubmitted manuscript.
- Why the focus in the Introduction section on Carbon dots, and what is its relationship with the present study?
Response: Carbon dots (CDs) are defined as carbonaceous nanoparticles that exhibit physical and optical properties analogous to conventional quantum dots and silicon nanoparticles. In this study, we fabricated the phytochemical-derived carbon dots (or nanoparticles) from one of the ginger non-volatile and phenolic components called zingerone. Zingerone NP has characterized its physicochemical properties are corresponding to the carbon dots, including the consist of the carbonaceous element, owns small size (<10 nm), change in UV-spectra absorption peak, and reconstitutes its molecular structure. Accordingly, zingerone NP can be regarded as the carbon dots. (Detail description of physicochemical properties has been included on page 9-10 with a yellow mark in the “discussion” section of the resubmitted manuscript. Also shown below, Question 3 "response".)
- Why the incubation time of 24h for the cytotoxicity assays? It would be interesting to study higher incubation times and see their potential effects on the cytotoxicity of the nanoparticles.
Response: A short-term treatment (i.e., 6 h treatment) of zingerone and/or zingerone NP would induce a limited effect on cell viability, even in the higher dose of 200 µM treatment the cells were still maintained an 80-100 % of cell viability. Moreover, the IC50 of both zingerone and zingerone NP at short-term treatment was also higher (over 400 and 330 µM, respectively) (Figure I). Therefore, a 24 h of incubated time was chosen for our subsequent experiments. Of course, by long-term incubation (7-10 days), zingerone NP treatment has been demonstrated its superior efficiency in anti-cell viability and toxicity by using a colony formation and colony survival assay ( the Figure 2 of the resubmitted manuscript).
Figure I. Cytotoxicity effects of Zingerone NPs at short time incubation of OSCC cell lines. Cells were treated with various doses of zingerone and/or zingerone NPs for 6 h. A short-term effect of zingerone and/or zingerone NP on cytotoxicity was analyzed using an MTT assay. This data indicated that doses lower than or equal to 200 µM of zingerone NPs simply induced limited cytotoxicity in these OSCC cell lines.
- While the studied nanoparticles were already reported, it would be beneficial for the readers if the authors describe them, their synthesis and properties in more detail in the present paper.
Response: Thanks for the reviewer’s suggestion. We have added the description of the zingerone NP synthetic detail and physicochemical properties in the resubmitted manuscript. As shown below:
The page 4 with yellow mark in the in the “Materials and Methods” section
2.1 Cell lines and nanosized zingerone
Three human oral squamous cell carcinoma (OSCC) cell lines including Ca9-22 ( the gingival SCC cells), Cal-27 and SAS ( the tongue SCC cells) ……... Zingerone was purchased from Sigma–Aldrich and used without any purification. Moreover, we have generated the zingerone NP using a hydrothermal methodology and described in our previous study.[29] Briefly, zingerone was dissolved in pure ethanol (2% w/v) as a stock concentrate of 100 mM. ……. and subjected to filtration using a 0.45 μm polyvinylidene fluoride (PVDF) syringe.
The page 9 -10 with yellow mark in the in the “Discussion” section
The as-prepared zingerone NP is an ultrasmall size of 1.42 ± 0.67 nm and owns a clear and amber apparent. Moreover, the zingerone NP possesses a negative charge (−15.99 ± 0.23 mV) and is observed an absorption peak at 350 nm of the UV-vis spectra due to π−π* transition of the benzenoid rings. The total ion chromatogram analysis also elucidates an exhibition of functional groups including methoxy and hydroxyl groups on the nanoparticle.……Indeed, the nanosized zingerone may own more surface chemistry exhibition of methoxy and hydroxyl groups rather than zingerone alone.[29]
- The authors should compare the cytotoxicity/IC50 values obtained for both zingerone and zingerone NPs for reference chemotherapeutic drugs for this type of cancer. Such information is essential to really understand the therapeutic potency of the reported nanoparticles.
Response: We appreciate the reviewer’s comments. We have collected previous studies and compared the IC50 of chemotherapeutic drugs such as cisplatin and 5-fluorouracil (5-FU) (which are the first-line drug for clinical oral cancer treatment.) and both zingerone and zingerone NP in these three OSCC cell lines. As shown in Table I, in general, as compared to chemotherapeutic drugs, both zingerone and zingerone NP have revealed a higher IC50 at 24 h treatment. Moreover, besides the SAS cell line (almost less study was reported the IC50 of 5-FU in SAS cell within 24 h treatment), Cal-27 cells have been observed a higher IC50 in 5-FU treatment. This result suggested that both zingerone and zingerone NP were more safe and biocompatible as compared to chemotherapeutic drugs. Accordingly, zingerone NP may be more adept for a phytochemical adjuvant for chemoprevention on oral cancer as compared to zingerone.
Table I. Comparison with the IC50 among zingerone, zingerone NP, and chemotherapeutic drugs in three OSCC cell lines.
- Use of adjectives as "excellent" is not very scientific, and so, the authors should refrain from using them.
Response: We appreciate the reviewer’s suggestion. We have changed the “excellent” into either “superior” or “good” in the resubmitted manuscript.
- The zingerone nanoparticles showed different activity than that of molecular zingerone. The authors should discuss in more detail potential reasons for that.
Response: Thanks for the reviewer’s suggestion. Besides the aspect of physicochemical properties in the resubmitted manuscript, we herein have added some sentences and discussed in more detail potential reasons for zingerone NP-elicited more bioactivity than the raw zingerone in the resubmitted manuscript. As shown below:
The page 10 with yellow mark in the in the “Discussion” section
“…..These outcomes may finally cause lethal stress for cancer cells. On the other hand, we have reported that zingerone NP significantly enhanced cell apoptosis, anti-cell proliferation, and disturbed cell cycle distribution in human hepatoma cells. …….. Accordingly, both Akt/NFkB signaling and caspases-cascade signaling might be involved in the zingerone NP -mediated anti-cell proliferation and anti-tumorigenicity in OSCC cell lines.”
- The authors should also discuss reasons for potential differences between obtained results for different cell lines.
Response: Thanks for the reviewer’s suggestion. We have discussed the reasons for potential differences between obtained results for different OSCC cell lines. As shown below:
The page 11 with yellow mark in the in the “Discussion” section
“Additionally, as compared to these tongue squamous cell carcinoma Cal-27 and SAS cell lines, we found that zingerone NP showed more bioactivity and tough suppression toward gingival squamous cell carcinoma Ca9-22 cells among MTT assay (Figure 1), colony formation (Figure 2), wound healing assay (Figure 3), and MMPs activity (Figure 6)……………... All the different genome backgrounds may contribute to the different tolerances during undergone zingerone and/or zingerone NP treatment. Moreover, the gene expression profiling and different gene heterogeneity in mankind races and sexual would also confer different sensitivity to zingerone and/or zingerone NP treatment.”
- The authors obtained results for carcinoma cells. To assess potential biocompatibility/safety issues, the authors should also evaluate the present nanoparticles toward non-carcinoma cells.
Response: Thanks for the reviewer’s comments. Because we have no normal oral epithelial cells, we herein used the immortalized human vascular endothelial cells (EA.hy926 cells) which were developed via fusion between primary human umbilical vein cells (HUVE cells) with lung cancer A549 cells. We detected the effect of both zingerone and zingerone NP on cytotoxicity of EA.hy926 cells at 24 h using MTT assay. As shown in Figure II, EA.hy926 cells were showed higher tolerance to zingerone NP treatment and obtained a higher IC50 (297.3±6.3 µM) as compared to the three OSCC cell lines (118.5±5.6 µM, 174.4±17.5 µM, and 147±17 µM in Ca9-22, Cal-27, and SAS, respectively). This data suggested that zingerone NP displayed the potential biocompatibility and safety toward non-carcinoma cells.
Figure II. The effect of Zingerone NP on cytotoxicity in human vascular endothelial cells EA.hy926 cells. Cells were treated with zingerone and/or zingerone NP for 24 h. The effects of zingerone and/or zingerone NP on cell viability was analyzed and recorded using a MTT assay.
Indeed, in our previous study, we have demonstrated the effects of both zingerone and zingerone NP on liver functions of animal models through detection of the food intake, liver weight and GOT and GPT indexes. As shown in Table I, our results indicated that both zingerone and zingerone NPs did not affect animal food uptake or the liver weight. Moreover, both zingerone and zingerone NPs significantly increased the feed conversion ratio (FCR) and stimulated the appetite of the mice thus leading to increased body weight. Furthermore, after animals were administered either zingerone or zingerone NPs for 8 weeks, the sera of GOT and GPT index were detected. Our result indicated that both GOT and GPT index showed no significant changes. These data suggest that the fabricated zingerone NPs are safe and have no liver toxicity. (These detail descriptions have been addressed in the Table S1 of “ACS applied materials & interfaces 11(1): 137-150 (2019)”.)
Table I. The effects of zingerone and zingerone NPs on mice food uptake, liver weight and liver function index in vivo.
Values are means ± SD for five mice in each group. FCR, Feed conversion ratio, indicates “the amount of feed needed for each additional gram of body weight”; GOT, Glutamate oxaloacetate transaminase; GPT, Glutamate pyruvate transaminase. *P < 0.05 and **P < 0.01. a: No significance.
- What is the synthesis yield (mass per mass, in %) of the obtained nanoparticles?
Response: We regarded the synthesis yield of zingerone NP as 100% due to in our synthesis process, we prepared zingerone (2% w/v) in EtOH as the stock solution A. On the other hand, zingerone NP solution was prepared from the same stock solution though further hemiketal reaction. This was used as zingerone NPs stock solution B. Later biological performance test was compared from these two stock solutions that were diluted to desired concentrations. We have described the concentrations accordingly in the material and methods section as shown below:
Page 4 with yellow mark of the “Materials and methods” section
2.1 Cell lines and nanosized zingerone
Three human oral squamous cell carcinoma (OSCC) cell lines including Ca9-22 ( the gingival SCC cells), Cal-27 and SAS ( the tongue SCC cells) ……... Zingerone was purchased from Sigma–Aldrich and used without any purification. Moreover, we have generated the zingerone NP using a hydrothermal methodology and described in our previous study.[29] Briefly, zingerone was dissolved in pure ethanol (2% w/v) as a stock concentrate of 100 mM. ……. and subjected to filtration using a 0.45 μm polyvinylidene fluoride (PVDF) syringe.
- How many replicate synthesis were performed for the nanoparticles, to ensure that they presented reproducible properties?
Response: The as-fabricated zingerone NP was synthesized through the one-pot self-assembled synthesis process. The zingerone NPs result from a hemiketal reaction between two zingerone compounds owing to the alcohol group condensation with the adjacent ketone group. This hemiketal reaction further polymerizes and finally exhibits a nanotetramer structure. moreover, besides the basic tests in physicochemical characterizations (as shown below, the response of Question 13 ), every batch fabricated zingerone NP will be detected the bioactivity using MTT assay in the same cell lines and that was all for making sure the as-fabricated zingerone NP is reproducible.
- To my understanding, the authors present these nanoparticles as carbon dot-like, which were obtained by bottom-up procedures. It should be noted that bottom-up synthesis of carbon dots produce also molecular impurities, besides the dots, which can only be removed from solution by either dialysis or column chromatography (see DOIs: 10.1021/acs.chemmater.7b04446; 10.1039/C9CP03730F). However, as I understood from their ACS Appl Mater Interface paper, the authors did neither. So, the authors should discuss how they guarantee the absence of potential molecular impurities.
Response: In our previous data, we did not observe the potential molecular impurities such as trimer formation or other molecules formed in our system. We mentioned in "Characterization and structure definition of zingerone NPs" of the Results and Discussion section, the zingerone NP constituents were directly observed by LC-MC analyses that found two peaks at 32.96 (peak I) and 33.77 min (peak II), respectively. By analyzing the ESI (+) MS/MS mode of zingerone NPs, the major peak of m/z 194 was consistent with zingerone (Figure III-D). To further verify the m/z values of peak I and peak II, we analyzed several major ion fragments and obtain their m/z values which respectively corresponded to 409.03, 216.95, 194.98 and 136.99 in peak I (Figure III-E) and 795.24, 409.03, 216.95, 194.98 and 136.99 in peak II (Figure III-F). Based on these results, we speculate that the constitutive structure of zingerone NPs results from a hemiketal reaction between two zingerone compounds owing to the alcohol group condensed with the adjacent ketone group. This hemiketal reaction further polymerizes and finally exhibits a nanotetramer structure. (ACS applied materials & interfaces 11(1): 137-150 (2019))
Figure III. Total ion chromatogram analysis for clarifying the constitutive structure of zingerone NPs. (This data was referenced from our previous study: ACS applied materials & interfaces 11(1): 137-150 (2019) )
- The authors should discuss how they can guarantee the uniformity of the population of obtained nanoparticles, without further purification.
Response: The zingerone NPs result from a hemiketal reaction between two zingerone compounds owing to the alcohol group condensation with the adjacent ketone group. This hemiketal reaction further polymerizes and finally exhibits a nanotetramer structure. To demonstrate the zingerone NP reproducibility, four batches of zingerone NPs, which were fabricated at different times, were subjected to several experiments including photograph images, TEM, and UV absorption spectra. The photographs show the zingerone NP resulting from four different batches with the same golden yellow color (Figure II). TEM images show similar zingerone nanoparticle formation (Figure III), and the particle sizes calculated are 1.42±0.67 nm, 1.67±0.89 nm, 2.07±1.35 nm and 1.78±1.27 nm, respectively. Moreover, the UV absorption peak of the zingerone NPs was found at 350 nm from four batches (Figure IV. batch 1, 2, 3 and 4). These results show this synthesis process is highly reproducible.
Figure IV. The photographs for zingerone NPs of batch 1, 2, 3 and 4.
Figure V. TEM images for zingerone NPs of batch 1, 2, 3 and 4. The particle sizes of zingerone NPs from each batch (a, b, c and d) by TEM images analysis are 1.42±0.67 nm, 1.67±0.89 nm, 2.20±1.85 nm and 1.78±1.27 nm, respectively.
Figure VI. UV absorption spectra of zingerone NPs from batch 1, 2, 3 and 4.

Round 2
Reviewer 1 Report
The authors have responded to all my questions. The manuscript is noticeably improved.
Author Response
We appreciate the reviewer’s kindly comment.
Reviewer 3 Report
The authors have addressed my comments, and so, my recommendation is for acceptance.
Author Response
We appreciate the reviewer’s comment. We have corrected all grammatical and typographical errors in the revised manuscript which were proofread by a professional editing team-American Journal Experts (AJE). The editorial certificate is shown below.